# Kawasaki Disease with Hepatobiliary Manifestations

**DOI:** 10.3390/medicina58121833

**Published:** 2022-12-12

**Authors:** Siti Aisyah Suhaini, Abdullah Harith Azidin, Chooi San Cheah, Wendy Lee Wei Li, Mohammad Shukri Khoo, Noor Akmal Shareela Ismail, Adli Ali

**Affiliations:** 1Department of Paediatric, Faculty of Medicine, Universiti Kebangsaan Malaysia, Jalan Yaacob Latif, Kuala Lumpur 56000, Malaysia; 2Department of Paediatric, Universiti Kebangsaan Malaysia Specialist Children’s Hospital (HPKK), Jalan Yaacob Latif, Bandar Tun Razak, Cheras, Kuala Lumpur 56000, Malaysia; 3Department of Biochemistry, Faculty of Medicine, Universiti Kebangsaan Malaysia, Jalan Yaacob Latif, Kuala Lumpur 56000, Malaysia

**Keywords:** hepatobiliary manifestation, Kawasaki disease, gallbladder hydrops, hepatitis, hepatomegaly, hyperbilirubinemia, complications

## Abstract

*Background and Objectives:* Kawasaki Disease (KD) incidence has been on the rise globally throughout the years, particularly in the Asia Pacific region. KD can be diagnosed based on several clinical criteria. Due to its systemic inflammatory nature, multi-organ involvement has been observed, making the diagnosis of KD more challenging. Notably, several studies have reported KD patients presenting with hepatobiliary abnormalities. Nonetheless, comprehensive data regarding the hepatobiliary manifestations of KD are limited in Malaysia, justifying a more in-depth study of the disease in this country. Thus, in this article, we aim to discuss KD patients in Malaysia with hepatobiliary manifestations. *Materials and Methods:* A total of six KD patients with hepatobiliary findings who presented at Hospital Canselor Tuanku Muhriz (HCTM) from 2004 to 2021 were selected and included. Variables including the initial presenting signs and symptoms, clinical progress, laboratory investigations such as liver function test (LFT), and ultrasound findings of hepatobiliary system were reviewed and analyzed. *Results:* Out of these six KD patients, there were two patients complicated with hepatitis and one patient with gallbladder hydrops. Different clinical features including jaundice (*n* = 3) and hepatomegaly (*n* = 4) were also observed. All patients received both aspirin and intravenous immunoglobulin (IVIG) as their first-line treatment and all of them responded well to IVIG. The majority of them (*n* = 5) had a complete recovery and did not have any cardiovascular and hepatobiliary sequelae. *Conclusions:* Despite KD mostly being diagnosed with the classical clinical criteria, patients with atypical presentations should always alert physicians of KD as one of the possible differential diagnoses. This study discovered that hepatobiliary manifestations in KD patients were not uncommon. More awareness on the epidemiology, diagnosis, and management of KD patients with hepatobiliary manifestations are required to allow for the initiation of prompt treatment, thus preventing further complications.

## 1. Introduction

Kawasaki disease (KD) is an acute systemic vasculitis that was first reported in 1961 [1]. Over the past few years, multiple papers have been published to provide a better insight of this disease. KD is characterized as an acute systemic vascular disease that mostly affects the small and medium vessels [2]. KD is self-limiting and happens commonly among children under 5 years old, following the diagnostic criteria in the American Heart Association (AHA) guidelines [3,4]. Most of the morbidity and mortality in KD patients stem from cardiac involvement with the development of arrhythmias or coronary artery aneurysms (CAA) [5]. However, in many patients, the clinical manifestations of KD are incomplete and atypical, which leads to delayed diagnosis and a worse prognosis for CAA. In this study, we refer to atypical presentations of KD as clinical presentations that were not listed as the classical manifestations under the AHA guideline [4]. Various atypical presentations of KD occur at an early age including hepatobiliary manifestations. Some KD patients presented initially with hepatobiliary manifestations, such as jaundice, abdominal pain, nausea, vomiting, diarrhea, hepatosplenomegaly, gallbladder hydrops, laboratory, and radiological hepatobiliary abnormalities, thus masking the classical symptoms of KD, leading to misdiagnosis of hepatobiliary or gastrointestinal system diseases such as hepatitis or acute acalculous cholecystitis (ACC) [6,7,8,9]. Although hepatobiliary manifestations do not belong to the classical criteria of KD, there were approximately 15% to 45% of patients who presented with these atypical presentations [10]. In terms of complications, CAA is most commonly reported, but the inflammatory lesions of KD are not only limited to the coronary arteries but can also involve the abdominal arteries [11]. It has been suggested that hepatic dysfunction has a relation with KD systemic inflammation; however, its nature is still not clearly understood [12]. Thus, patients with atypical hepatobiliary manifestation should raise the index of suspicion of KD as one of the differential diagnoses. Since there is a scarcity of studies on KD in Malaysia, this study aims at discussing KD patients presenting with hepatobiliary manifestations at our center to further aid in the understanding of KD clinical presentations.

## 2. Material and Methods

### 2.1. Study Location and Period

The search for KD cases with hepatobiliary manifestation was conducted at Hospital Canselor Tuanku Muhriz (HCTM), a tertiary medical center and a teaching hospital under the administration of Universiti Kebangsaan Malaysia (UKM). The study was conducted from October 2021 to October 2022. Ethical approval was obtained from the institute itself (HCTM) prior to the commencement of this study (JEP-2021-868).

### 2.2. Research Design

The study was conducted as a retrospective cohort study. A total of 103 patients who attended HCTM with the diagnosis of KD from 2004 to 2021 were initially retrieved from the HCTM Case Mix system by using the International Classification of Diseases (ICD) code, ICD-10 (M30.3), for mucocutaneous lymph node syndrome, which is another name for KD. From this cohort, a total of 6 KD cases with hepatobiliary manifestations, such as jaundice, hepatomegaly, hepatitis, or gallbladder hydrops, were identified and included in this study. Variables including the initial presenting signs and symptoms, clinical progress, laboratory investigations such as liver function tests, and ultrasound findings of the hepatobiliary system were reviewed and analyzed.

### 2.3. Inclusion and Exclusion Criteria

All registered data of patients who were admitted to HCTM between 2004 and 2021 with the diagnosis of KD based on the health information system were retrieved. With this approach, there were two methodological limitations. The first limitation was the type I error, which happened when non-KD patients were coded as KD in the system. These patients did not meet the criteria to be diagnosed as complete or incomplete KD and were excluded from this study. The second limitation was the type II error, which happened when KD patients were not coded as KD in the system. These patients’ data could not be traced and subsequently were not included in this study. From the total of 103 KD patients’ data that were retrieved, those who had hepatobiliary manifestations, such as jaundice, hepatomegaly, hepatitis, or gallbladder hydrops, were identified and included in this study.

Meanwhile, the exclusion criteria of this study are (i) patient data with repeated names and reference numbers and (ii) patient data that could not be accessed at all due to loss of information or the patient’s file. Repetitions of data were considered as a single entry. However, any incomplete dataset was accepted and reported as it is. After considering all the inclusion and exclusion criteria, the total of this study’s subject is 6.

## 3. Results

### 3.1. Hepatitis

Out of the six KD patients, there were two patients diagnosed with hepatitis simultaneously with KD. Both patients presented with jaundice and abnormal liver function test. Patient 1 had incomplete KD, and serological investigation for hepatitis A immunoglobulin G (Ig G) tested positive, which led to the delayed diagnosis of KD. Meanwhile, Patient 2 had hepatomegaly with the typical presentations of KD, making an earlier diagnosis of KD complicated with hepatitis.

Patient 1, with underlying glucose-6-phosphate dehydrogenase (G6PD) deficiency, presented with fever and cough for 3 days. Acute gastroenteritis (AGE) was diagnosed by a general practitioner and symptomatic treatment was given. On the next day, rashes started to develop over the chest, trunk, and upper limbs. Jaundice was also noted. Upon admission on day 4 of illness, examination revealed bilateral cervical and inguinal lymphadenopathy, injected throat, cracked red lips, desquamation of scrotal area, and mild hepatomegaly. Laboratory investigations showed neutrophilic leukocytosis (white cell count 16.4 × 10^9^ L; neutrophil 92%), direct hyperbilirubinemia (total serum bilirubin 142 µmol/L; direct bilirubin 111 µmol/L), and elevated alanine aminotransferase (ALT: 75 U/L). The patient was initially managed as viral hepatitis with concurrent G6PD hyperbilirubinemia. Serological investigations showed positive for Hepatitis A immunoglobulin G (IgG), indicating a previous history of infection. Kawasaki disease (KD) with atypical presentation was only diagnosed after the onset of edema and widespread rash over the extremities.

Patient 2 was initially treated for tonsilitis and received antibiotics. However, the fever (average recorded temperature of 39 °C) and left-sided neck pain persisted. Referral to our hospital for further evaluation was only done on the 7th day of fever with rashes over the back, neck, and cubital and popliteal fossa; bilateral non-purulent conjunctivitis; unilateral lymphadenopathy; and jaundice. Upon admission, the patient’s liver was 3 cm palpable below the right subcostal margin, indicating the presence of hepatomegaly. Other systemic examinations were unremarkable. Laboratory studies revealed leukocytosis (white cell count 29.7 × 10^9^/L), hemoglobin of 11.1 g/dL, normal platelet count (230 × 10^9^/L) and raised C-reactive protein (CRP) (30 mg/dL). Serum alkaline phosphatase (ALP: 447 U/L) and serum ALT (82 U/L) were elevated. Diagnosis of typical KD with mild hepatitis was considered, and echocardiographic (ECHO) examination was performed revealing a normal cardiac finding.

In terms of management, both patients received intravenous immunoglobulin (IVIG) 2 g/kg with high-dose aspirin therapy followed by subsequent low-dose aspirin therapy. Both patients were subsequently on follow-ups at cardiology clinic and were later discharged with no complications.

### 3.2. Gallbladder Hydrops

There was one KD patient within our cohort with the finding of gallbladder hydrops. Patient 3 presented with fever for 8 days with the highest temperature of 39 °C associated with rigors and rashes. Rashes started to develop on day 4 of illness and initially appeared around the perioral region, then radiated to the ear, scalp, trunk, and limbs within a few hours. However, these rashes subsided on day 6 of illness. Upon admission, there were red lips and tongue, bilateral conjunctival injection, left axillary lymphadenopathy, and flaring of the bacille Calmette-Guérin scar (BCGitis), hence the diagnosis of complete KD. Examinations of other systems were unremarkable. Laboratory investigations showed leukocytosis (16.3 × 10^9^ L, neutrophils 4.7 × 10^9^) and hypoalbuminemia with normal ALT and ALP levels. Ultrasound of the abdomen was done, and gallbladder hydrops was confirmed, in which the gallbladder wall was distended without debris measured 4.9 cm in length (normal pediatric gallbladder measurement for 0–1-year-old, length of gallbladder range between 1.3 and 3.4 cm). The echocardiography result was normal. Treatment with IVIG of 2 g/kg over 12 h and oral aspirin of 30 mg/kg/day for 6 days was given. The patient was then continued with low-dose aspirin of 4 mg/kg/day for 6 weeks and subsequently discharged with no complications.

### 3.3. Hepatomegaly

Hepatomegaly is one of the hepatobiliary manifestations in KD. Four out of six patients in this study were found to have hepatomegaly findings. Two of the patients (Patient 1 and Patient 2) had concurrent hepatitis and we described the patients in Section 3.1. Under this subsection, we will focus on the two cases presented with hepatomegaly with normal LFT results.

The first patient, Patient 4, presented with 9 days of non-resolving low-grade fever (38 °C), which was temporarily relieved by tepid sponging. Maculopapular rashes developed on day 2 of illness and spread from both lower limbs towards the upper limbs, trunk, and face. Lips were dry; however, no classic KD mucosal lesions were noted. On systemic examination, other systems were normal, except the liver was 2 cm palpable (hepatomegaly). Blood investigations showed raised white cell count (47.2 × 10^9^ with neutrophilia) and elevated CRP (20.78 mg/dL) with normal liver function test. The patient was tested to be rotavirus positive and was diagnosed with acute gastroenteritis (AGE). Ultrasound of the abdomen showed hepatomegaly with non-specific pericholecystic fluid. After 12 h of admission, pustular lesions developed and were noted to worsen especially around the thigh region, and the patient was started on intravenous antibiotics. On day 3 of admission, the patient was noted to have redness of both eyes and peeling of periungual region on his back eventually spreading to the abdomen and right upper limb. With this constellation of signs, the patient was eventually diagnosed with typical KD.

The second patient, Patient 5, presented with prolonged cough for 2 months and high-grade fever for 5 days associated with maculopapular rash that started on the trunk and abdomen, then later became generalized, while only sparing the face. There were also eye redness and reduced oral intake. On examination, there were generalized maculopapular rash, BCGitis, and hepatomegaly with the liver palpable 2 cm below the right subcostal margin. Otherwise, systemic examinations were normal. Laboratory investigations revealed high CRP (10.52 mg/dL). Incomplete KD was subsequently considered, and treatment was initiated.

In terms of management, both patients were treated well with IVIG at 2 g/kg over 12 h and high-dose oral aspirin. Echocardiography results of Patient 4 revealed right coronary artery ectasia of a 4 mm diameter, and the patient was prescribed low-dose aspirin (5 mg/kg/day) for 2 months. Meanwhile, Patient 5′s echocardiography result was normal, and the patient was only given low-dose aspirin (5 mg/kg/day) for 6 weeks.

### 3.4. Cholestatic Jaundice

Our study found that cholestatic jaundice can also be one of the hepatobiliary manifestations of KD. Patient 6, with underlying G6PD deficiency, presented with 5 days history of fever associated with swollen lips and tongue, bilateral non-purulent conjunctivitis, and BCGitis with generalized macular rashes, as well as 1 day history of jaundice with passing of tea-colored urine. On examination, the patient appeared fretful and had jaundice with red and dry crack lips, generalized macular rash, and flaring of BCG scar. Cervical lymph nodes were palpable with the biggest measuring 1 cm × 1.5 cm. The throat was injected, and tonsils were enlarged. Systemic examinations were unremarkable.

Laboratory investigations revealed leukocytosis (white cell count 21.5 × 10^9^/L) with neutrophilia. His CRP was elevated at 28.94 mg/dL. The total bilirubin was 142 µmol/L with the direct component of 112.8 µmol/L, indicating direct hyperbilirubinemia. The patient also had high levels of ALP (451 U/L), ALT (139 U/L), lactate dehydrogenase (LDH: 407 mmol/L), and Gamma-glutamyl transferase (GGT; 338 U/L). However, ultrasound abdomen revealed no evidence of gallbladder hydrops.

The patient was treated with IVIG2 g/kg over 16 h. After completion of IVIG, the patient remained afebrile. Oral aspirin of 30 mg/kg/day was given for 6 days and later tapered down to 4 mg/kg/day for 6 weeks. Upon discharge, the patient was well with improving liver function and no other complications noted.

## 4. Discussion

The cohort of KD patients reported in this series highlights the importance of the high level of suspicion that is required for KD to be diagnosed accurately, especially if patients come in with the atypical hepatobiliary presentations (Table 1).

The clinical manifestations of KD can be diverse. The diagnosis of KD can be difficult as not all the clinical features appear simultaneously. Hematological and biochemical investigations are not immensely helpful; however, these could exclude other diagnosis. Moreover, establishing the diagnosis of KD can be further complicated by the occurrence of other diseases, such as hepatitis and gallbladder hydrops, as seen in our cases. Our study reported six KD patients with hepatobiliary system manifestations, of which two had hepatitis, one had gallbladder hydrops, four had hepatomegaly, and three had jaundice, with one case manifested as cholestatic jaundice. All these patients presented with hepatobiliary manifestation simultaneously with the appearance of KD features, which further made the diagnosis of KD more challenging. In some cases, we reported a misdiagnosis in the first phase of the disease, causing delayed diagnosis of KD and late definitive treatment to be offered to the patients. This was similarly reported in previous study, where there was a delayed diagnosis of KD due to the initial misdiagnosis of viral hepatitis [13].

There were four KD patients noted to have hepatomegaly, which is an uncommon feature in KD. This was suggested by the possible involvement of portal area inflammation during acute phase of KD [14]. This clearly showed that hepatobiliary manifestations can affect the judgement of physicians to diagnose KD especially when patients had prominent hepatobiliary symptoms. Undeniably, hepatobiliary manifestations were widely reported as one of the clinical features of KD; however, they are not included in the classical clinical criteria [12,13,15].

We reported several KD patients with jaundice but without gallbladder hydrops, which was also observed by Taddio et.al [16]. This has been found to be a rarer occurrence, in which patients who presented with clinical jaundice had no sonographic evidence of gallbladder hydrops or mechanical obstruction [17,18,19,20,21]. In other studies, KD patients presented with obstructive jaundice were later found to develop gallbladder hydrops with symptoms mimicking acute abdomen [22]. One of the possible explanations behind this occurrence is lymphadenopathy causing compression effect of the hilum of the liver (porta hepatis) [7].

Apart from atypical clinical manifestations of KD, laboratory and radiological investigations demonstrating hepatobiliary abnormalities could also assist in the confirmation of the diagnosis. Three patients were reported to have abnormal LFT with either elevated ALT, ALP, or both. The pathogenesis of LFT derangement in KD is incompletely understood but is thought to be multifactorial [12]. Proposed etiologies include generalized inflammation, vasculitis, congestive cardiac failure secondary to myocarditis, non-steroidal anti-inflammatory antipyretics, toxin-mediated effects, or a combination of these conditions [12]. Liver dysfunction in KD patients is usually self-limiting, and the median recovery time ranges from 2 days to 99 days [23,24]. Although LFT is not the diagnostic tool of KD, it may indicate the severity of ongoing inflammation, thus serving as a prognostic marker for the development of IVIG resistance or coronary artery aneurysm (CAA) [12,15,25]. Having said that, patients can also present with normal LFT, as ALT was only elevated in less than 40% of KD patients and hyperbilirubinemia only occur in 10% of KD patients. Meanwhile, hypoalbuminemia was common among patients with severe and prolonged KD [4].

Based on our study, although hepatobiliary manifestation is not one of the criteria to diagnose KD, it is important to remember such unusual presentations do not exclude KD. Diagnosing KD in those who presented with atypical presentations of KD remains a challenge for physicians. Undeniably, one of the reasons for delayed diagnosis of KD is due to the atypical presentations [26]. Investigations, discussion with experts, and review of published guideline are mandatory to confirm or exclude the diagnosis of KD [27]. Delayed diagnosis causing delayed treatment will lead to increased risk for CAA to develop [28]. Therefore, a good and broad awareness of the various KD presentations is of utmost important to avoid delayed in the diagnosis, so that prompt treatment can be achieved.

Being known as an X-linked recessive genetic disorder, certain variants of G6PD deficiency including Class I, II, and III will cause potentially life-threatening hemolytic anemia with exposure of triggers, such as infection, drugs, and fava beans [29]. To date, the relationship between aspirin treatment and hemolytic anemia in KD patients with G6PD remains unknown due to limited studies [30]. Two of our KD patients within the studied cohort had underlying G6PD deficiency. However, both patients had direct hyperbilirubinemia during the acute phase of KD, indicating hepatocellular injury and negating the possibility of hemolytic anemia secondary to G6PD deficiency, which further suggests that aspirin treatment is not absolutely contraindicated in KD patients with underlying G6PD deficiency.

One of the major limitations of our study is the loss of patient’s data since this is a retrospective analysis of KD patients in HCTM. Therefore, some of the information, such as laboratory investigation and duration of follow-up and subsequent treatment, was missing and could not be reported completely.

## 5. Conclusions

In this study, we discussed six KD patients who presented with hepatobiliary manifestations. Some of them initially had atypical presentations of KD, while others were misdiagnosed with other diseases before KD was considered. Therefore, abnormal results from laboratory investigations, such as LFT and imaging study including ultrasound of the hepatobiliary system, should always raise a suspicion of KD in patients who fulfilled only some of the classical clinical features of KD. KD with unusual presentations require for extreme alertness, rapid diagnosis, and prompt treatment to prevent progression to coronary artery lesions or potentially life-threatening disease with severe long-term consequences.

## Figures and Tables

**Table 1 medicina-58-01833-t001:** Summary of the Clinical Hepatobiliary Presentation and Management of the KD patients.

	Patient 1	Patient 2	Patient 3	Patient 4	Patient 5	Patient 6
Age (months)	24	72	6	4	4	22
Duration of fever (days)	3	7	8	9	5	5
Highest temperature (°C)	40	39	39	38	39	39
Classical features of KD	Incomplete	Complete	Complete	Complete	Incomplete	Complete
Bilateral non-purulent conjunctivitis	-	+	+	+	+	+
Extremity changes	-	+	+	+	+	-
Maculopapular rash	+	+	+	+	+	+
Oral mucosal changes	+	-	+	-	-	+
Cervical lymphadenopathy	+	+	+	+	-	+
BCGitis	-	-	+	-	+	+
URTI symptoms	+	+	-	+	+	-
GIT symptoms	+	-	-	+	-	-
Jaundice	+	+	-	-	-	+
Hepatomegaly	+	+	-	+	+	-
Gallbladder hydrops	-	-	+	-	-	-
CRP (mg/dL)	12.30	30.00	7.14	20.78	10.52	28.94
ALT (U/L)	75	82	43	17	46	136
ALP (U/L)	264	447	143	99	232	497
GGT (U/L)	NA	NA	NA	NA	NA	338
Bilirubin (µmol/L)	142.0	96.1	3.9	6.0	6.2	149.9
Treatment	IVIG 2 g/kg over 12 h Aspirin 30 mg/kg/day for 2 weeks followed by 5 mg/kg/day for 3 months	IVIG 2 g/kg over 16 h Aspirin 30 mg/kg/day for 5 days followed by 5 mg/kg/day for 6 weeks	IVIG 2 g/kg over 12 h Aspirin 30 mg/kg/day for 6 days followed by 4 mg/kg/day for 6 weeks	IVIG 2 g/kg over 12 h Aspirin 30 mg/kg/day for6 days followed by 5 mg/kg/day for 2 months	IVIG 2 g/kg over 12 h Aspirin 30 mg/kg/day for 3 days followed by 5 mg/kg/day for 6 weeks	IVIG 2 g/kg over 16 h Aspirin 30 mg/kg/day for 6 days followed by 4 mg/kg/day for 6 weeks.
Complications	No	No	No	CAA	No	No

+ Present; - Absent; NA Not available; CAA Coronary artery aneurysm.

## Data Availability

Not applicable.

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
