# Peer review of "Kawasaki Disease with Hepatobiliary Manifestations"

_medicina, 2022, doi:10.3390/medicina58121833_

Round 1
Reviewer 1 Report
The KD with atypical presentation always is challenging. It is very important to make diagnosis early due to the need of specialized treatment. The article presents cases of KD with hepatobilary manifestations, so is actual for general practitioners and pediatricians. But I have some remarks:
1. I suggest to expand the table 1, or to make new table, including the age of the patient, clinical features and laboratory investigations, time (day) of treatment. It will make the information more clear.
2. Follow-up time. Were your patients monitored or not for liver function? If yes, how long? What data?
3. Aspirin therapy. What doses of aspirin were prescribed in 2 patients with hepatitis and in second patient with hepatomegaly? How long your patients received high doses of aspirin? Why the third patient (with gall bladder hydrops) was prescribed high dose of aspirin 30 mg/kg for a long time- 6 weeks?
Author Response
Please see the attachment, thank you.

Reviewer 2 Report
Reviewer comments
Title
KAWASAKI DISEASE WITH HEPATOBILIARY MANIFESTATIONS
The authors wrote an interesting article about hepatobiliary manifestations of Kawaski Disease. The manuscript is well written .However ,some important points need to be addressed .
· 2
Title: no need to write all letters in capital
· 19
reported their KD patients. English editing is required in the whole manuscript
· 22
Materials and Methods: A total of 6
The number of patients is too low .So ,we can consider this manuscript a case series not a research article .
· 29
(n=4,67%)
When your sample size is n <20, give the actual numbers, and no percentages.
· 44
papers were published to have a better understanding
English editing is mandatory
· 46
It is self-limiting and normally happens among children under 46 5 years old which is diagnosed according to the diagnostic criteria in the American Heart As-47 sociation (AHA) guidelines
Too long distance and difficult to be understood
· 64
The research gap and aim of the study should be written at the end of the introduction section
· 67
atypical hepatobiliary
what is meant by the word atypical .Please give a precise definition
· 72
The study design was not defined
The author did not mention the approval of the study ..where
· 65
The writing of the materials and methods section is immature .the authors should define the inclusion and exclusion criteria precisely
· 85
positive serological investigation
English editing is mandatory
· 87
together with the typical presentations of KD
English editing is mandatory
· 141
Hepatomegaly is one of the hepatobiliary manifestations in KD as out of the 6 KD pa-141 tients there were 3 patients found to have hepatomegaly findings.
The structure of some sentences needs to be revised and edited
218
with hepatobiliary manifestation
English editing
· The limitations of the study were not mentioned

Author Response
Please see the attachment, thank you.

Round 2
Reviewer 1 Report
The Results chapter must be improved: the data in the text and table should not repeat itself. If all the data are included in the text , the table is unneccesary. But the data would be more clear if the clinical symptoms of KD and hepatobilary symptoms as well as all laboratory tests would be included in the table.
References
I noticed that newest sources about KD were not mentioned in the article, for example, Diagnosis, Treatment, and Long Term Management of KD, Circulation, Volume 135, Issue 17, Apr 2017
European consensus-based recommendations for the diagnosis and treatment of KD- the SHARE initiative by de Graeff and al., 2019
and others....
P.S. in two references No 9 and No 22 there is no year mentioned, 46 percent of sources are older then 10 years
Author Response
Please see the attachment, thank you.

Reviewer 2 Report
The manuscript improved greatly after addressing of the comments
Author Response
We would like to thank and express our appreciation to the reviewer for spending the time to improve our manuscript.
